# Endogenous Chemerin from PVAT Amplifies Electrical Field-Stimulated Arterial Contraction: Use of the Chemerin Knockout Rat

**DOI:** 10.3390/ijms21176392

**Published:** 2020-09-02

**Authors:** Emma D. Flood, Stephanie W. Watts

**Affiliations:** Department of Pharmacology and Toxicology, Michigan State University, 1355 Bogue Street Rm B445, East Lansing, MI 48824, USA; dariosem@msu.edu

**Keywords:** chemerin, chemerin KO rat, electrical field stimulation, PVAT

## Abstract

**Background:** We previously reported that the adipokine chemerin, when added exogenously to the isolated rat mesenteric artery, amplified electrical field-stimulated (EFS) contraction. The Chemerin1 antagonist CCX832 alone inhibited EFS-induced contraction in tissues with but not without perivascular adipose tissue (PVAT). These data suggested indirectly that chemerin itself, presumably from the PVAT, facilitated EFS-induced contraction. We created the chemerin KO rat and now test the focused hypothesis that endogenous chemerin amplifies EFS-induced arterial contraction. **Methods:** The superior mesenteric artery +PVAT from global chemerin WT and KO female rats, with endothelium and sympathetic nerve intact, were mounted into isolated tissue baths for isometric and EFS-induced contraction. **Results:** CCX832 reduced EFS (2–20 Hz)-induced contraction in tissues from the WT but not KO rats. Consistent with this finding, the magnitude of EFS-induced contraction was lower in the tissues from the KO vs. WT rats, yet the maximum response to the adrenergic stimulus PE was not different among all tissues. **Conclusion:** These studies support that endogenous chemerin modifies sympathetic nerve-mediated contraction through Chemerin1, an important finding relative in understanding chemerin’s role in control of blood pressure.

## 1. Introduction

Epidemiological evidence strongly supports the positive association between circulating levels of the secreted protein and adipokine chemerin [1,2], body mass index (BMI) and blood pressure [3,4,5]. Chemerin is secreted primarily by the liver and fat depots [6,7,8]. The actions of chemerin as an adipokine implicate this relatively new protein in dysfunctions of clinical diseases such as obesity, given that fat is a significant source of chemerin. Chemerin’s expression in perivascular adipose tissue (PVAT) of rat and human [8,9] is of particular interest because it has the immediate potential to directly modify blood pressure. We previously determined two important means by which chemerin could increase blood pressure through vascular mechanisms.

First, activation of the primarily biological receptor for chemerin, Chemerin1 (also known as ChemR23 or chemokine like receptor 1 /CMKLR1) [10], with the agonist chemerin-9 [11] caused direct and receptor-dependent arterial contraction [8]. Second, exogenous chemerin-9 amplified the sympathetic nerve-mediated electrical field-stimulated (EFS) contraction of the isolated mesenteric artery [12]. In the same EFS experiment, we discovered that the Chemerin1 antagonist CCX832 [12] alone reduced EFS-induced contraction in arteries +PVAT but not in arteries –PVAT. Thus, PVAT provides a substance that stimulates Chemerin1 to promote contraction. While we knew chemerin was expressed in PVAT [13], we had, at that time, no way to prove definitively that the protein chemerin, itself, was this substance. These data only supported than an activated Chemerin1 supported EFS-induced contraction. This has been a critical gap in our knowledge in determining whether reducing chemerin expression, antagonizing Chemerin1 or both could be potential therapeutic approaches. We recently created the chemerin KO rat through CRISPR-Cas technology [14] and here use tissues from this rat to test the specific hypothesis that it is chemerin that is necessary for amplifying EFS-induced contraction. Our model was the superior mesenteric artery, given our reported findings.

## 2. Results

Using Western analyses, we verified that chemerin protein in the plasma was lost in the KO but present in the WT females that were used in this experiment (Figure 1A). Protein was present in the lanes of the KO samples, as indicated by the total protein stain (bottom blot, Figure 1A). These data were densitized and quantified in Figure 1B. Chemerin was abolished in the KO vs. the WT samples. This is consistent with previous findings in which plasma chemerin is absent because tissue (adipose, liver) chemerin is absent [14].

Arteries from chemerin WT and KO rats all contracted to the α_1_ adrenergic agonist phenylephrine with statistically similar magnitudes (in milligrams: WT Vehicle = 892 ± 113; WT CCX832 = 877 ± 57; KO Vehicle = 757 ± 30; KO CCX832 = 846 ± 60; *p* > 0.05). Similarly, the magnitude of 20 Hz-stimulated initial contraction was not different when comparing tissues from the WT and KO rats (in milligrams; WT Vehicle = 522 ± 61; WT CCX832 = 496 ± 34; KO Vehicle = 487 ± 47; KO CCX832 = 494 ± 24; *p* > 0.05). TTX virtually abolished the 20 Hz-induced contraction, to 1.56–3.74% of initial 20 Hz contraction, indicating that the EFS-induced response was nerve mediated. We have previously determined this response was also α_1_ adrenergic receptor dependent (e.g., antagonized by the antagonist prazosin), indicating that EFS-induced contraction is driven by sympathetic nerve activation [12]. Thus, vessels from WT and KO rats have similar contractile potential.

Isolated arteries were incubated with vehicle or the Chemerin1 antagonist CCX832 (100 nM) for 30 min prior to EF stimulation. CCX832 consistently reduced EFS-induced contraction in tissues from WT (Figure 2A) but not KO rats (Figure 2B). CCX832 elevated EFS-induced contraction at 10 and 20 Hz stimulus in tissues from the KO rat. Moreover, the vehicle EFS-induced contraction was significantly lower in the KO vs. the WT, suggesting that the KO tissue had lost a substance that normally contributed to EFS-induced contraction.

## 3. Discussion

This work fills an important gap in our knowledge. These data strongly support that endogenous chemerin, likely coming from the PVAT, positively amplifies the actions of the sympathetic nerve and thereby increases vascular tone. Though a straightforward finding, the implications are novel, significant and important. When one combines this new knowledge with the information that chemerin also directly contracts the isolated artery, chemerin becomes a potentially formidable agent in control of total peripheral resistance and blood pressure. This is evidenced by the recent finding that chemerin infusion in the mouse elevates blood pressure [15]. Moreover, the ability of chemerin to potentiate the effects of the activated sympathetic nerve has now been observed in both the male [12] and female vasculature (this work). This is relevant given the important findings that the relationship between sympathetic nerve activity in the muscle of the women does not exist as it does in men [16]. Does such a difference exist throughout the sympathetic nervous system (e.g., outside of skeletal muscle)? Such studies would illuminate whether the role of chemerin as described in the present study is different between male and female and/or changes in the face of disease.

An important outcome from this work is to emphasize that adipokines, including chemerin, are molecules that have multiple biological functions, not just those functions for which they were originally discovered. The chemerin gene, known as either tazarotene-induced gene 2 (*TIG2*) or retinoic acid receptor responder 2 (*RARRES2)*, was discovered in psoriatic skin lesions [17]. Over the course of 10 years, studies revealed that chemerin is also a chemoattractant [3,18]. In 2007, chemerin was first described as an adipokine [6,7,8]. Chemerin joins adipokines such as leptin in its ability to amplify the effects of sympathetic nerve function [19]. Chemerin1 was colocalized with tyrosine hydroxylase, a sympathetic nerve marker, in the isolated rat superior mesenteric artery [12], but the precise mechanisms by which activation of this receptor modifies nerve function are not known. Other adipokines thought to work in concert/potentiate the effects of sympathetic nerve function include aldosterone and neprilysin [20], resistin [21] and angiotensin II [22]. 

Another outcome is the importance of understanding more about the actions of CCX832, one of the few chemerin receptor inhibitors available. CCX832 alone enhanced EFS-induced contraction at the 10 and 20 Hz stimuli in tissues from the KO rat. CCX832 does not act as an adrenergic receptor antagonist [12]. There is also no evidence that CCX832 acts as anything other than a neutral antagonist, but it is a relatively new tool. We cannot exclude the possibility that CCX832 acts as a weak agonist to promote EFS-induced contraction done in the absence of chemerin. Data such as these point out the clear need to study the pharmacology of CCX832 more deeply. We also recognize a limitation of results from the 20 Hz stimulus in these experiments. While the original 20 Hz stimulus, given directly after the initial PE challenge, tended to be lower in the KO tissues vs. the WT, this was not statistically significant. That contrasts with the significant differences between the WT and KO response to the 20 Hz stimulus in the formal series of EFS challenges. We can only speculate that the tissue has better equilibrated at this point in the protocol; we will be mindful of what this potential means.

On a larger scale, this work is relevant when one considers obesity-associated hypertension, in which chemerin burden would be greater because of excess adipose tissue that includes PVAT. Hypertension and obesity are the primary conditions that contribute to cardiovascular disease [23,24,25]. These two diseases are inextricably linked in the USA, with nearly 70% of adults overweight or obese and ½ of adults are hypertensive. Obesity contributes to up to 70% of essential hypertension [26]. We have not been able to prevent or treat obesity-associated hypertension and there is no specific treatment for this disease. As such, new solutions, such as the antagonism of chemerin production or function, are needed. This work provides such direction.

## 4. Materials and Methods 

### 4.1. Animals

The Michigan State University Institutional Animal Care and Use Committee (IACUC) approved all protocols (Protocol # 02-18-026, approved Feb 18 1018). Chemerin WT and KO rats were bred from three separate sets of chemerin heterozygous parents, as previously described [14]. All rats were genotyped prior to use, and presence (WT) or lack (KO) of chemerin verified experimentally (described in [14]). Only females were used in this study because a sufficient number of paired WT and KO male rats were not born from the litters generated. Rats were not staged.

### 4.2. Materials

CCX832 was provided by ChemoCentryx (Mountain View, CA USA). Dimethylsulfoxide, phenylephrine hydrochloride and prazosin were purchased from Sigma Chemical Company (St. Louis, MO USA). Tetrodotoxin was purchased from Tocris (Minneapolis, MN USA).

### 4.3. Tissue Preparation and Contractility

Rats were anesthetized with pentobarbital (60–80 mg/kg i.p.) and the superior mesenteric artery +PVAT was removed and placed in physiological salt solution (PSS) containing (mM): NaCl 130; KCl 4.7; KH_2_PO_4_ 1.18; MgSO_4_ × 7H_2_O 1.7; NaHCO_3_ 14.8; dextrose 5.5; CaNa_2_EDTA 0.03, CaCl_2_ 1.6 (pH 7.2). Sections of the artery +PVAT were cut into 3–4 mm rings and mounted between two stainless steel hooks. Only sections +PVAT were used, as EFS-induced contraction is not potentiated in the absence of PVAT. One hook was fixed within the warmed (37 °C) and aerated (95% O_2_, 5% CO_2_) tissue bath (30 mL). Two platinum electrodes positioned within the tissue bath were placed around the tissue, with electrodes connected to a Grass Instruments stimulator model S88 (Quincy MA). The other hook was connected to a Grass isometric force transducer (FT03; Grass Instruments, Quincy, MA, USA) connected to an ADInstruments PowerLab (ADInstruments, Colorado Springs, CO, USA). Tissues were placed under optimal resting tension (1200 mg) and equilibrated 1 h before an initial challenge with a maximal concentration of phenylephrine (PE; 10^−5^ M). Tissues were then washed until tone returned to baseline.

An initial 20 Hz maximal stimulus was delivered to each vascular segment to validate the presence of a functioning nerve. The 20 Hz contraction was ~50% of the PE response. The fast sodium channel inhibitor tetrodotoxin (TTX; 300 nM) was then added to the tissue bath and, after 30 min, the 20 Hz stimulus repeated to validate the neuronal origin of the stimulus. Tissues were washed and then EFS stimuli (30 stimuli, stimulus duration 0.5 ms, frequency 0.3–20 Hz, voltage 14 V) were delivered sequentially from low to high stimulation, with at least 20 min of washing. CCX832 (100 nM) or vehicle (0.01% DMSO) was added for a 30-min incubation, after washing, prior to each stimulus.

### 4.4. Western Analyses

Plasma was used to validate chemerin presence in the WT and loss in the KO rats. One hundred micrograms total protein derived from each animal were loaded per lane on standard polyacrylamide gels (15%). Gel proteins were transferred to a PVDF-FL membrane using wet transfer. A LI-COR REVERT® kit (Lincoln, NE, USA) was used to stain total protein and scanned on 700 channel of a LICOR Odyssey CLx (Omaha Nebraska, USA). The membrane was then rehydrated in methanol, rinsed in water, and total protein stain removed with Reversal solution for 5 min, then rinsed twice. All blots were then blocked in 4% chick egg ovalbulmin (catalog #A5378, Sigma, Sigma Chemical Company, St. Louis, MO, USA) for three hours on a rocker at 4 °C. Chemerin antibody (catalog #112520, Abcam, Cambridge, MA USA; RRID:AB_10864055) diluted to 1:1000 in blocking buffer was incubated for 48–72 h with constant rocking at 4 °C. Blots were then rinsed with tris-buffered saline + 0.1% Tween-20 three times for 10 min, followed by incubation with LICOR IRDye secondary antibody (catalog # 926-32210, 800CW goat anti-mouse at 1:1000, Lincoln, NE, USA; RRID:AB_621842) for an hour with rocking. Secondary solution was removed, blot was washed 3 × 10 min with tris-buffered saline + Tween, and imaged on a LICOR Odyssey CLX in the 800 channel. 

### 4.5. Data Analyses

All quantified results are presented as the means ± SEM. Images from Western analyses are presented as a whole, and brightness/contrast modified on the image as a whole, not in part. Signals from the chemerin bands are normalized to the total protein stain for that entire tissue lane. Contraction is reported as a percentage of an original 20 Hz stimulated contraction or in milligrams. A one-way ANOVA was used to determine differences in maximum contraction to PE. A two-way ANOVA with a Bonferroni correction was used to determine statistical differences in EFS maximums. In all cases, *p* < 0.05 was considered significant.

## Figures and Tables

**Figure 1 ijms-21-06392-f001:**
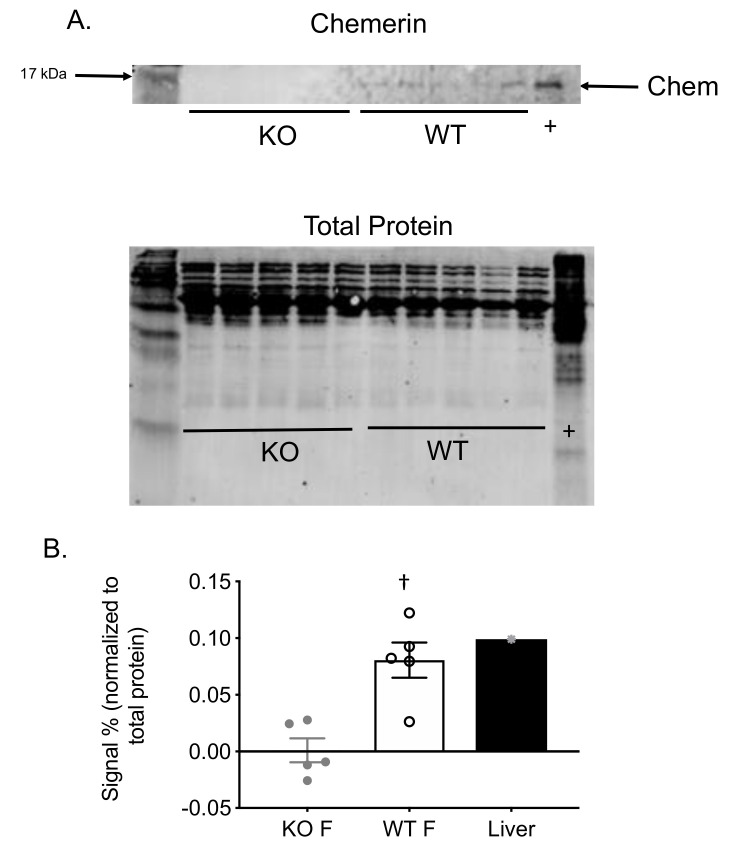
Chemerin protein was absent in KO plasma. (**A)**: Blot showing actual experiment of chemerin expression in the plasma of the wild type female (WT F) but not the knockout female (KO F) rats, with each lane representing a separate animal. Liver homogenate (Liver) from a normal male Sprague–Dawley rat was used as a positive control for chemerin detection. Total protein stain of the blot prior to incubation with chemerin antibody is shown below the chemerin blot. (**B)**: Densitometry of chemerin signal as a percentage of the total protein stain normalized for each lane. Bars represent means ± SEM for number of lanes scattered around the mean. † = statistical difference from paired WT response using an unpaired Student’s t test.

**Figure 2 ijms-21-06392-f002:**
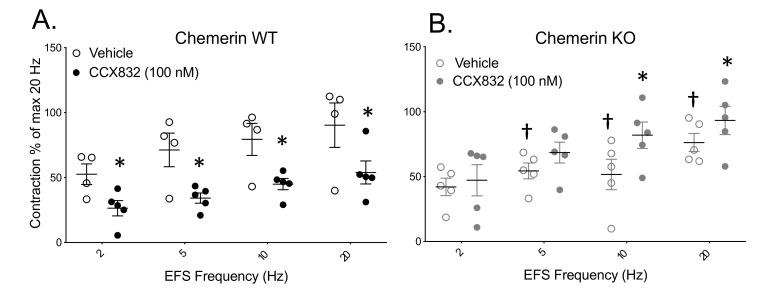
Endogenous activation of Chemerin1 by chemerin amplifies EFS-induced contraction in WT but not KO tissues. The Chemerin1 antagonist CCX832 alone reduced EFS-induced contraction in the superior mesenteric arteries from WT (**A**) but not KO rats (**B**). Horizontal bars represent means ± SEM for number of points scattered around mean. A two-way ANOVA with a Bonferroni correction was used to determine statistical differences in EFS maximums. * = statistical difference from paired vehicle response; † from frequency paired WT response.

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
