# Peer review of "Endogenous Chemerin from PVAT Amplifies Electrical Field-Stimulated Arterial Contraction: Use of the Chemerin Knockout Rat"

_ijms, 2020, doi:10.3390/ijms21176392_

Round 1

Reviewer 1 Report

Very well written and presented. 

Author Response

Thank you for taking the time to read our manuscript.  We DO know this is a short report, but felt this was an essential piece of information to implicate a particular adipokine - chemerin- in an important event, namely modifying sympathetic nerve function.

We are grateful to you.

Reviewer 2 Report

Using chemerin-KO rats, the authors (Emma Flood and Stephanie Watts) show that this adipokine, present among others in perivascular tissue, potentiates sympathetic nerve-mediated arterial contraction and thus could be involved in the control of vascular tone. This study is novel and original but some results which were not discussed need more explanations.Additionally, the text needs English revision and corrections. 

Major comments: 

1-  Figure 1B: What do KO F and WT F mean?  What is the column reffered as ‘liver’? Is it a positive control? No mention was made for that in the text. What are the chemerin levels in PVAT of those WT and KO animals? 

2-  Initial EFS- and PE-induced contractions should be shown in a figure.

3-  Figure 2: CCX832 reduced EFS-induced contractions in WT mesenteric artery whatever the frequency. In KO rats, at 10 and 20 Hz, CCX832 potentiated contractile response when compare to paired vehicle response. What is the significance of that observation and how do the authors explain that effect? To complete the study, the effect of exogenous chemerin should be studied on arteries in the same experiments.  In the first part of the results section, the authors wrote that no difference was observed between WT and KO for the 20Hz-stimulated initial contraction. In figure 2, we can see that not only the 20Hz-induced contraction but also other EFS-induced contraction (except for 2Hz) are statistically different between WT and KO. What is the explanation of that difference? This needs to be discussed.   

Minor comments:

1- lines 58-59: correct the sentence

2- line 75: Isolated arteries were incubated with vehicle or with the chemerin1 receptor antagonist

3- line 76: CCX832 consistently reduced EFS-induced contraction instead of CCX832 reduced EFS-induced contraction consistently

4-lines 94-97: What do the authors mean by these two sentences? This sounds contradictory and not self-consistent. Indeed, if the ability of chemerin to potentiate the effects of sympathetic nerve activation is observed both in male and in female and if the relationship between sympathetic nerve activity and obesity is different between male and female, this should imply the non-involvement of chemerin in this phenomenon.

5-Line 170: prefer ‘as a percentage of the initial 20Hz-stimulated contraction’ instead of ‘the original contraction’.

6-References 1 and 2 do not have the same format as the other references.

Author Response

Thank you for helping us to improve the clarify of this manuscript.  We respond here to you and with requested revisions in the revised manuscript.  Your criticisms are in BOLD font, while ours are in normal font.  In the revised manuscript, changes made are in a red font. 

Major comments: 

1-  Figure 1B: What do KO F and WT F mean?  What is the column reffered as ‘liver’? Is it a positive control? No mention was made for that in the text. What are the chemerin levels in PVAT of those WT and KO animals? 

This is our fault for not defining these better in the figure legend; this has now been done and we are sorry we did not do this in the first place.  The revised figure legend reads:

Figure 1. Chemerin protein was absent in KO plasma.  A: Blot showing actual experiment of chemerin expression in the plasma of the wild type female (WT F) but not the knockout female (KO F) rats, with each lane representing a separate animal.  Liver homogenate (Liver) from a normal male Sprague Dawley rat was used as a positive control for chemerin detection.   Total protein stain of the blot prior to incubation with chemerin antibody is shown below the chemerin blot.  B.  Densitometry of chemerin signal as a percentage of the total protein stain normalized for each lane.  Bars represent means+SEM for number of lanes scattered around the mean. † = statistical difference from paired WT response.

The chemerin level in the PVAT of the WT and KO animals has been published (Reference 14).  It is completely absent in the KO vs the WT, in which chemerin can be detected readily. We made the following statement in the original manuscript:

This is consistent with previous findings in which plasma chemerin is absent because tissue (adipose, liver) chemerin is absent [14].    

2-  Initial EFS- and PE-induced contractions should be shown in a figure.

As a short communications, we have been encouraged to not use more than two figures, hence reporting of these important values are present in the manuscript, just within the text.  We hope this is acceptable. 

3-  Figure 2: CCX832 reduced EFS-induced contractions in WT mesenteric artery whatever the frequency. In KO rats, at 10 and 20 Hz, CCX832 potentiated contractile response when compare to paired vehicle response. What is the significance of that observation and how do the authors explain that effect?

This is interesting - we hadn't thought of this this way.  There is no evidence that CCX832 acts as anything other than a neutral antagonist, but it is a relatively new tool.  It could be that CCX832 acts as an weak agonist to promote EFS-induced contraction, though the mechanism just isn't known.  We now state this possibility in the revised manuscript results, and follow in the discussion with this potential explanation: 

Results: CCX832 elevated EFS-induced contraction at 10 and 20 Hz stimulus in tissues from the KO rat.

Discussion:  Another outcome is the importance for of understanding more about the actions of CCX832, one of the few chemerin receptor inhibitors available.  CCX832 alone enhanced EFS-induced contraction at the 10 and 20 Hz stimuli in tissues from the KO rat.  There is no evidence that CCX832 acts as anything other than a neutral antagonist, but it is a relatively new tool.  We cannot exclude the possibility that CCX832 acts as an weak agonist to promote EFS-induced contraction.  Data such as these point out the clear need to study the pharmacology of CCX832 more deeply.    

To complete the study, the effect of exogenous chemerin should be studied on arteries in the same experiments.  In the first part of the results section, the authors wrote that no difference was observed between WT and KO for the 20Hz-stimulated initial contraction. In figure 2, we can see that not only the 20Hz-induced contraction but also other EFS-induced contraction (except for 2Hz) are statistically different between WT and KO. What is the explanation of that difference? This needs to be discussed. 

This work, at least in the normal rat that is the same strain (Sprague Dawley) upon which the chemerin KO was built, has already been published by our lab.  This is reference 13 in the manuscript.  It was, in fact, these studies that demonstrated that exogenous chemerin could amplify EFS-induced contraction that led us to examine whether endogenous chemerin that could potentiate EFS-induced contraction.

We appreciate the point that adding back chemerin to reverse the loss in EFS-induced contraction in the KO tissues would be interesting, and we considered these experiments originally.  We decided, however, not to do them for the following reason.  We'd have to do this experiment in both the WT and KO, and chemerin will potentiate contraction in both groups; this experiment is uninterpretable in being to say with precision that chemerin 'filled in the hole' in the KO vs WT.  There is also the matter of concentration, of what to add back to the tissue, as what comes from the tissue and gets to the nerve/artery is not something we know.   As such, we considered these experiments to muddy the water more and thus did not do them.  We hope this makes sense.  

Minor comments:

1- lines 58-59: correct the sentence; 

This sentence is complete; we are unsure what you are looking for.  There is a period, and this is a complete sentence.

2- line 75: Isolated arteries were incubated with vehicle or with the chemerin1 receptor antagonist

We have revised this to be consistent with our use of Chemerin1 for chemerin1 receptor not only here but throughout the manuscript; this is nomenclature we still have not adapted to!

3- line 76: CCX832 consistently reduced EFS-induced contraction instead of CCX832 reduced EFS-induced contraction consistently

Corrected as suggested and now found on line 81. 

4-lines 94-97: What do the authors mean by these two sentences? This sounds contradictory and not self-consistent. Indeed, if the ability of chemerin to potentiate the effects of sympathetic nerve activation is observed both in male and in female and if the relationship between sympathetic nerve activity and obesity is different between male and female, this should imply the non-involvement of chemerin in this phenomenon.

I think we see what you mean, and we've added to the paragraph with the following sentences (line 104):

Does such a difference exist throughout the sympathetic nervous system (e.g. outside of skeletal muscle)?  Such studies would illuminate whether the role of chemerin as described in the present study is different between male and female and/or changes in the face of disease.   

5-Line 170: prefer ‘as a percentage of the initial 20Hz-stimulated contraction’ instead of ‘the original contraction’.

Your suggestion is precisely what was stated on what is now line 193 of the revised manuscript.

6-References 1 and 2 do not have the same format as the other references.

Thank you - we missed the bolding in these references, and it is now present in these specific references.

Reviewer 3 Report

The subject of the article new and important, once the work show that endogenous chemerin modifies sympathetic nerve-mediated contraction through Chemerin 1, an important finding relative in understanding chemerin’s role in control of blood pressure.

General comments

  • The Abstract is clear and well structured.
  • In the introduction, pleased discuss the clinical importance of the work.
  • In the results included the statically test used in the figure legends.
  • In the results the authors say that similar magnitudes were obtain in the contractions with phenylephrine and with 20 Hz-stimulated contractions. In the iscussion the authors say that “endogenous chemerin, likely coming from the PVAT, positively amplifies the actions of the sympathetic nerve and thereby increases vascular tone.” 
  • Should the effect of phenylephrine be diminished on Chemerin KO? pleased explain the data.
  • The discussion is very good, clear and well structured and the clinical importance of the work is well addressed.

Author Response

Thank you for helping us improve this work.  We know this is a hard time to ask individuals for their time, and thus this is appreciated.

Your comments are in bold font, and our response is in the normal font that follows.  You will see our response as action with the revised submission in red font. 

  • In the introduction, pleased discuss the clinical importance of the work.

We have added a sentence to the Introduction (line 32):

The actions of chemerin as an adipokine implicate this relatively new protein in dysfunctions of clinical diseases such as obesity, given that fat is a significant source of chemerin

That links to what we say in the final paragraph of the discussion, sharing WHY we think this work is clinically relevant. 

  • In the results included the statically test used in the figure legends.

Statistical tests are now specified in legends for figures 1 and 2.

  • In the results the authors say that similar magnitudes were obtain in the contractions with phenylephrine and with 20 Hz-stimulated contractions. In the iscussion the authors say that “endogenous chemerin, likely coming from the PVAT, positively amplifies the actions of the sympathetic nerve and thereby increases vascular tone.” 

I think I see what you mean - how can the initial 20Hz contraction not be different between the WT and KO, but the 20 Hz contraction in the formal experiment be different?  By sheer magnitude, the 20 Hz-induced contraction in the KO was lower than the WT (now lines 63-64), but the variability in those initial contractions was not statistically significant.  We've added this important limitation to the revised discussion (beginning line 125). 

We also recognize a limitation in results the 20 Hz stimulus in these tissues.  While the original 20 Hz stimulus, given directly after the initial PE challenge, tended to be lower in the KO tissues vs the WT, this was not statistically significant.  That contrasts with the significant differences between the WT and KO responses in the 20 Hz stimulus of the formal series of EFS challenges.  We can only speculate that the tissue has better equilibrated at this point in the protocol; we will be mindful of what this potential means.     

  • Should the effect of phenylephrine be diminished on Chemerin KO? pleased explain the data.

This is a good question, given that we know phenylephrine can potentiate the effects of chemerin.  In cited reference 13, we published that chemerin given exogenously (1 uM) cannot change the potency or maximal contraction to norepinephrine (NE), nor can CCX832 change these same parameters.  This suggests that CCX832 is not interfering with adrenergic receptor stimulation.  There is so much more we could say, but we've added the following simple sentence to the revision to clarify the most important idea (line 121). 

CCX832 does not act as an adrenergic receptor antagonist [13].

  • The discussion is very good, clear and well structured and the clinical importance of the work is well addressed.

Thank you. 

Round 2

Reviewer 2 Report

No additional comments.

The authors have answered to all points.

Reviewer 3 Report

This article was revised appropriately.

I recommend accept